# Near-Isogenic Lines of *Japonica* Rice Revealed New QTLs for Cold Tolerance at Booting Stage

**Zhenghai Sun** [1,2,3,4,*,†], **Juan Du** [5,†], **Xiaoying Pu** [5,†], **Muhammad Kazim Ali** [5,6,7,†], **Xiaomeng Yang** [5], **Chengli Duan** [2,8], **Meirong Ren** [1], **Xia Li** [5] and **Yawen Zeng** [5,*] 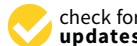

[1]   School of Horticulture and Gardening, Southwest Forestry University, Kunming 650224, China; xiaohei504@hotmail.com

[2]   College of Agronomy and Biotechnology, Yunnan Agricultural University, Kunming 650201, China; chengliduan@hotmail.com

[3]   Yunnan Province South and Southeast Asia Joint R&D Center of Economic Forest Full Industry, Kunming 650224, China

[4]   Yunnan Province International Technological Cooperation Base of High Effective Economic Forestry Cultivation, Kunming 650224, China

[5]   Biotechnology and Germplasm Resources Institute, Yunnan Academy of Agricultural Sciences/Agricultural Biotechnology Key Laboratory of Yunnan Province, Kunming 650205, China; coockoo1976@126.com (J.D.); puxiaoying@163.com (X.P.); ali.kazimm@gmai.com (M.K.A.); yxm89ccf@126.com (X.Y.); lixia_napus@163.com (X.L.)

[6]   Department of Biotechnology, Federal Urdu University of Arts, Science and Technology, Karachi 75300, Pakistan

[7]   Karachi Institute of Biotechnology and Genetic Engineering (KIBGE), University of Karachi, Karachi 75270, Pakistan

[8]   School of Forestry and Biotechnology, Zhejiang Forestry University, Hangzhou 311300, China

*   Correspondence: sunzhenghai1978@163.com (Z.S.); zengyw1967@126.com (Y.Z.); Tel.: +86-871-658-94145 (Y.Z.)

†   These authors contributed equally to this work.

**Abstract:** Low temperature stress severely hampers rice productivity, and hence elaborating chilling-mediated physiochemical alterations and unravelling cold tolerance pathways will facilitate cold resilient rice breeding. Various cold tolerant Near-isogenic lines (NILs) selected at the booting stage through backcrossing of a *japonica* landrace Lijing2 (cold tolerant) with cold sensitive Towada (a *japonica* cultivar). The cold tolerance attributes of NILs was validated over two years by evaluating the spikelet fertility followed by correlation of nineteen morphological traits with the rate of seed setting (RSS). Results revealed BG, FG, 1-2IL, RSLL, and UIL were significantly correlated with RSS and had nearer marker interval distance with cold tolerance in QTL analysis. Two QTLs, *qCTB-7-a* and *qCTB-7-b*, were found for RSS based on a mixed linear model. Alleles of two QTLs were contributed by Lijing2 and genetic distances between the peaks were 0.00 and 0.06cM, which explained 5.70% and 8.36% variation, respectively, One QTL for 1-2IL, RSLL, and ILBS, while two QTLs for FG, BG, and UIL were also identified. These findings can be exploited to engineer low temperature stress tolerant rice in times of climate change.

**Keywords:** booting stage; rate of seed setting; cold tolerance; QTLs; NILs; rice

---

## 1. Introduction

Plant metabolic processes and life functions are used to work in optimum condition of prevailing environment, i.e., optimum temperature and light, but diversions from the optimum condition imparts severe damages in a plants life, owing to excessive production and accumulation of reactive oxygen species (ROS), obstruction in the process of photosynthesis, disturbance of ion and nutrient uptake,

and ultimately declines plant growth and yield [1–3]. Cereals are more prone to environmental clues and they experience diverse environmental conditions during their growth cycle. Low temperature stress, categorized as freezing (<0°C) and chilling (0–15 °C) stress, is the paramount milieu element restricting the topographical dispensation of crops and their growth and yield [4]. Hence, cold stress severely influences the growth and productivity of rice (*Oryza sativa* L.) and affects its distribution around the globe [5,6]. However, 4.5 million hectares of rice are grown per year all over the world including tropical and subtropical areas, especially in some areas of moderate climatic zones, which experience unpredicted chilling stresses [7,8]. Rice is one of the most important cereals and the major staple food crop consumed by nearly half of the world population and it constitutes about 21% of human nutrient intake and energy requirement. Particularly, in Asia, rice processed products provide 60%–70% daily calories for over two billion people [9,10].

Despite various strategies, adopted to develop cold tolerant rice cultivars, the unexpected chilling snaps caused 5%–10% reduction in yield annually and occasionally it could go up to 20%–40% [8,11]. In 24 rice producing countries, including the major areas of Japan, Korea and China, there are about 15 million hectares rice fields under threat of low temperature stress where yield losses are estimated about at three–five million tons per year [12–14]. Chilling stress causes severe damages at all rice phenology, but the flowering stage is more prone to extreme temperatures [5,6]. Many rice varieties cannot be planted in approximately seven million hectares of land in South-East and South Asia, due to the experience of chilling stress [5,15]. Chilling stress affects rice growth from germination to maturity and its severity depends on the phenological stage and duration of stress. Therefore, improving rice chilling stress tolerance through modern cultivation and breeding techniques is detrimental and crucial for sustainable agriculture and food security in these areas [5,16,17].

At booting stage, young microspores from the tetrad stage to the first contraction phase are more prone to chilling stress and are more critical for rice grain development [8,18–21]. The chilling stress at the booting stage maximized spikelet sterility and caused minimum grain yields [22,23]. The failure of the developing putative pollen grains in the microspore development stage is more responsible for spikelet sterility under low temperatures stress condition [5,6,24,25]. Therefore, to identify chilling stress tolerant rice genotypes, spikelet sterility is used to determine the most common trait at this stage [26,27].

Previously, several quantitative trait loci (QTLs) have been identified using backcross population, doubled-haploid lines, or the F2 population, but due to their complex nature of genetic design, very few genes have been isolated and cloned [28,29]. Many morphological markers are correlated with low temperature stress tolerance including reciprocal second leaf length (RSLL), the first and second internode length (1-2IL), and anther length, however, the correlations between these traits with cold tolerance are not fully understood when the temperature descend at the booting stage. QTLs for morphological traits associated with chilling stress tolerance and QTLs controlling other key regulators for low temperature stress tolerance are not defined and mapped at present [26,30]. Chilling stress tolerance at this stage are those quantitative traits that are governed by multiple genes and partial QTLs. Thus, QTLs for cold tolerance are mapped at the booting stage distributed among 12 rice chromosomes. Among them, chromosome number four and five have maximum QTLs (eleven approximately) and chromosome number three has the least two for cold tolerance at the booting stage. Mapping populations for these QTLs include $F_{2:3:4}$, RILs, $BC_5F_5$, DHLs, and NILs. More QTLs for cold tolerance at the booting stage are propitious to clone and utilize genes for cold tolerance [5,22]. The molecular mechanism of cold stress tolerance at booting stage of rice development is still enigmatic and requires detailed study [5,22,26]. Thus, it is of a great importance and applications for future crop improvement, to screen for cold tolerant lines at this stage [31,32], which will pave the way to understand the genetic basis and molecular mechanisms of this great agronomic trait [12,33–35].

In China, Yunnan Province is one of biggest places in the world with diverse genotypes of *Oryza sativa* [12,36,37]. In the development of chilling stress tolerant rice varieties Yunnan province is of great importance with complicated geographical and climatic conditions situated on the Southwest

plateau. Among other rice cultivars, Lijing2 is considered as a cold tolerant rice variety at the booting and flowering stages, produced under such conditions [33,38].

Elucidation of life saving pathways for cold tolerance through identification of putative QTLs in Near-isogenic lines (NILs) will greatly benefit, but is a challenging task of agricultural industry [31]. In this study, a set of cold tolerant NILs was developed by backcrossing a rice variety Towada (Chilling sensitive) with Lijing2 (Chilling tolerant), to identify major QTLs for low temperature stress tolerance and subsequent cloning the gene, and to develop cold tolerant rice at the booting stage [39].

## 2. Materials and Methods

This study was carried out at two locations, namely experimental farm of Yunnan Academy of Agricultural Sciences, Kunming, China, and the second in a mountainous village of Aziying located 40 km away from Kunming, China, having 1916 m and 2150 m altitude, respectively. Some details of the experimental condition including geographical locations, temperature, planting period, and cold treatment are presented (Table 1). Plants subjected to low temperature stress (16–19 °C) under natural low atmospheric temperature when it reached between the differentiation of young panicles to the milky mature stage. The cold tolerant NILs progenies was obtained and selected in each backcross generation after successful crossing of cold tolerant rice cultivar (Lijing2) with cold sensitive rice cultivar (Towada). One of the cold tolerant lines $BC_4F_5$ (Towada//5/Lijing2) was chosen to cross with Towada to construct a segregating $BC_5F_3$ population and subsequent studies. Growth attributes and yield of $BC_5F_3$ population (264) and the parent (Towada and Lijing2) under cold stress were critically evaluated and the results were compiled.

**Table 1.** Experimental design and field conditions in this study.

|  | Experimental Locations | |
|---|---|---|
|  | **Kunming** | **Aziying** |
| Number of lines | 264 × 2 | 264 × 2 |
| Sowing date | Mid-March | Mid-March |
| Transplant date | Late May | Late May |
| Harvest date | Early October | Early October |
| Average temperature | 16–18 °C | 15–18°C |
| Cold treatment method | Natural low temperature | Cool-water irrigation system |
| Duration of booting to maturity | July to August | July to August |
| Air temperature (booting stage) | 19.0 °C | 17 °C |
| Water temperature | 18.5 ± 0.50 | 17.4 ± 0.30 |

### 2.1. Morphological Traits

Rice adaptation under cold stress, particularly at the flowering stage, was assessed through number of plant growth attributes specific to booting stage [8,22,27]. Therefore, rice growth attributes including anther length (AL), anther width (AW), plant height (PHt), effective tillering (ET), panicle length (PaL), flag leaf length (FLL), flag leaf width (FLW), uppermost leaf length (ULL), reciprocal first leaf length (RFLL), reciprocal first leaf width (RFLW), reciprocal secondary leaf length (RSLL), reciprocal secondary leaf width (RSLW), internode length below spike (ILBS), uppermost internode length (UIL), second internode length (SIL), first and second internode length (1-2IL), spike length (SL), full grains (FG), blighted grains (BG), and number of grain per panicle (NOGP) recorded during this study. Three individual plants per line in every repetition were noted and the mean of the two repetition (including six individual plants) as corresponding morphological traits. Full grains (FG), blighted grains (BG), number of grain per-panicle (NOGP = FG + BG. RSS = FG/NOGP × 100%), anther length (AL), and anther width (AW) were obtained in lab. Three inflorescences of individual plants per line in every repetition were measured by the Universal Projection meter and the mean of two repetitions (including eighteen inflorescence), as the corresponding anther was evaluated.

The correlations between RSS with nineteen other morphological traits were determined by using statistical software SPSS 20.0 (SPSS Inc., Chicago, IL, USA).

## 2.2. Extraction and Amplification of DNA

Extraction of genomic DNA was carried out by CTAB method using young and fresh leaves [40]. Concentration and purity of extracted DNA was determined by using Nanodrop. About 15 μL reaction mixture was prepared by the addition of 3 μL of PCR buffer (20 mM Tris (pH 8.0), 50 mM KCl, 2.5 mM $MgCl_2$, 0.1 mM EDTA, 1mM DTT, 50% glycerol), 3 μl genomic DNA (50 ng) and 2 μl (200 nM) of each primers (F and R), followed by the addition of 0.5 μL (0.1 mM) of each dNTPs and approximately 0.5 units of Taq polymerase (Beijing Dingguo Changsheng Biotechnology Co. Ltd. Beijing China). GeneAmp PCR System (Applied Biosystems, Foster City, CA, USA) were used to prepare 96-well amplification plate and the reaction condition of PCR amplification was set at 94 °C for 5 min followed by 34 cycles of the reaction, set at 94 °C, 55 °C, and 72 °C for 40, 40, and 60 s, respectively. Later final extension step was set at 72 °C for 5 min. The PCR products were loaded on 8% denatured polyacrylamide gels prepared by using DNA (vertical) electrophoresis unit and followed by electrophoresis at 80 Volts using 0.5 × TBE as running buffer. Silver-staining [41] was used to stain the gel and SSR marker bands were scored by GelScan for genome mapping and QTL analysis [42].

## 2.3. Construction of DNA Bulks and Molecular Marker Analysis

Bulk segregation analysis (BSA) and subsequent analysis of molecular markers, in 264 $BC_5F_3$ plants including NILs and parents, carried out as previously reported by Xu et al. [23]. Four DNA bulks were made, consisting of the first and second five most (or least) cold tolerant individuals for each phenotype followed by BSA analysis with SSR markers. To determine polymorphisms among the cold tolerant and cold sensitive NILs pools and parents (Lijing2, and Towada), a total of 600 SSR (Supplementary Table S1) markers covering all over 12 chromosomes of rice, which have 1440.5 cM of the entire genome and 2.4 cM for the mean marker interval (International Rice Genome Sequencing Project, 2005) were used. The RM-series primers of SSR markers were designed using previously reported methods [43–45]. PCR products (DNA bands) was assumed to probably linked with cold tolerance having the same band size as the cold tolerant bulks Lijing2, but different from the cold sensitive bulks of Towada. These SSR markers were further evaluated and processed for ANOVA.

## 2.4. Mapping and QTL Analysis

Each phenotypic and molecular data subjected to analysis of variance (ANOVA) using SPSS for Windows, version 20.0 (SPSS Inc., Chicago, IL, USA), to determine their significance in the interpretation of results. Kosambi function was used to convert recombination frequencies into genetic distances, as well as for linkage analysis using Mapmaker/EXP3.0 [46]. Composite interval mapping (CIM) was performed based on mixed linear model using QTL mapper V2.0 [47] because, QTL mapper V2.0, software can make unbiased estimates of QTL additive and dominance effects as compared to QTL mapper V1.0 [48,49]. For additive QTLs, analysis $p = 0.001$ was used as the probability threshold and a LOD score of 3.0 was used as another threshold to declare the presence of putative QTLs. The additive and dominance effects were estimated based on the percentage variation explained by each QTL.

Two linkage groups with 10 SSR markers have been constructed by MAPMAKER3.0 software at the max distance 50.0 range. A molecular linkage map has been drawn by the MapDraw2.1 (Huazhong Agricultural University, Wuhan, China) programmer, while QTL identification was carried out by using QTLMapper1.6 (Zhejiang University, Zhejiang, China), with a mixed linear model.

## 3. Results

### *3.1. Growth Attributes and Cold Stress Tolerance*

3.1.1. Correlations between Morphological Traits and RSS

Rate of seed setting (RSS) is the most exact index to evaluate rice cold tolerance at the booting stage, therefore rice growth attributes, correlated with RSS, could indirectly evaluate rice cold tolerance at booting stage. Eleven and thirteen morphological traits of rice, grown in Kunming and Aziying experimental locations, correlated with RSS as well as with nineteen other morphological traits, respectively, at 0.05 or 0.01 level (Table 2). From this, we conjectured that lower temperature at booting stage could affect more morphological traits; a similar conclusion has been drawn previously by Cui et al. [26].

**Table 2.** Correlation between morphological attributes including, Anther Length (AL), Anther Width (AW), Plant Height (PHt), Effective Tillering (ET), Peduncle Length (PL), Flag Leaf Length (FLL), Flag Leaf Width (FLW), Internode Length Below Spike (ILBS), Reciprocal First Leaf Length (RFLL), Reciprocal First Leaf Width (RFLW), Uppermost Internode Length (UIL), Reciprocal Secondary Leaf Length (RSLL), Reciprocal Secondary Leaf Width (RSLW), Second Internode Length (SIL), 1st and 2nd Internode Length (1-2IL), Panicle Length (PaL), Full Grains (FG), Blighted Grains (BG), and Number of Grain Per Panicle (NOGP), with cold stress tolerance indicator, rate of seed setting (RSS), of NILs population grown at Kunming (K), and Aziying (A).

| Morphological Traits | Rate of Seed Setting (RSS) | | | |
| --- | --- | --- | --- | --- |
| | **K** | **A** | **Mean** | **Difference** |
| AL | 0.19 ** | 0.24 ** | 0.23 ** | 0.02 |
| AW | 0.12 * | 0.17 ** | 0.13 * | 0.07 |
| PHt | 0.06 | 0.14* | 0.09 | 0.04 |
| ET | −0.17 ** | 0.08 | −0.06 | −0.14 * |
| PL | 0.07 | 0.03 | 0.02 | 0.10 |
| FLL | −0.08 | 0.00 | −0.05 | −0.05 |
| FLW | −0.32 ** | 0.29 ** | 0.00 | −0.18 ** |
| ILBS | −0.04 | 0.08 | −0.08 | 0.05 |
| RFLL | −0.11 | −0.17 ** | −0.06 | −0.16 ** |
| RFLW | −0.24 ** | 0.35 ** | 0.02 | −0.04 |
| UIL | 0.27 ** | 0.22 ** | 0.24 ** | 0.20 ** |
| RSLL | −0.23 ** | −0.12 * | −0.17 ** | −0.13 * |
| RSLW | −0.16 ** | 0.31 ** | −0.07 | 0.07 |
| SIL | −0.10 | 0.019 | −0.02 | −0.15 * |
| 1-2IL | 0.23 ** | 0.53 ** | 0.32 ** | 0.16 * |
| PaL | −0.05 | −0.17 ** | −0.07 | −0.03 |
| FG | 0.89 ** | 0.98 ** | 0.89 ** | 0.89 ** |
| BG | −0.91 ** | −0.12 * | −0.66 ** | −0.66 ** |
| NOGP | 0.098 | 0.057 | 0.00 | 0.204 ** |

Note: "*" and "**" means, significance at 0.05 and 0.01 level.

Low temperature increases the proportion of morphological traits correlated positively with RSS (45.5% and 69.2% morphological traits correlated positively with RSS in Kunming, and Aziying, respectively). However, some variations also observed, like FLW, RFLW, and RSLW, that attributes had negative correlations with RSS in Kunming, but it showed positive correlation in Aziying. All these traits are associated with rice width, so it may hypothesize that lower temperature effects more morphological traits that correlated with width. Among nineteen morphological traits, ten attributes i.e., AL, AW, FLW, RFLW, ULL, RSLW, RSLL, 1-2IL, FG, and BG were correlated with RSS at 0.05 or 0.01 level in both experimental locations, and therefore it can be concluded that these traits had a primary association with rice cold tolerance at booting stage. Prevenient research validated our experimental result by others experimental material [12,26,43]. Here we defined the correlation of ten morphological traits with cold tolerance as Class 1. Means of twenty morphological traits in

both experimental locations and correlations between them were calculated to elucidate the relation between rice cold tolerance and morphological traits at booting stage. The result showed seven traits (AL, AW, ULL, 1-2IL, RSLL, FG, and BG) out of nineteen morphological traits correlated with RSS at 0.05 or 0.01 level and these seven traits also correlated with RSS in both experimental locations. Seven traits including AL, AW, ULL, 1-2IL, RSLL, FG, and BG, correlated with rice cold tolerance at booting stage closer than those afore-mentioned ten traits, and hence, these seven traits are categorized as class 2 based on their correlation analysis.

Morphological traits of Towada NILs population neared to stabilization by self-cross time after time, however, the differences among them are mostly dependent on prevailing environment and phenological stage, and therefore correlation between rate of seed setting (RSS) with morphological traits except the temperature factor to explain if the different correlation between morphological traits with RSS was raised by temperature or not. Comparing the two locations for correlation analysis revealed ten morphological traits correlation with RSS at 0.05 or 0.01 level, however, five traits (ULL, 1-2IL, RSLL, FG, and BG) showed the closest correlation with rice cold tolerance at booting stage, and therefore these five morphological traits are grouped into Class 3 for rice cold tolerance. In five morphological traits, four traits correlated positively with RSS except BG, and therefore further analyses were carried out to determine the relationship between these five traits and molecular markers followed by QTL analysis.

### 3.1.2. Variation of Six Morphological Traits Correlated with Rice Cold tolerance at Booting Stage

The variation of six morphological traits (Table 3 and Figure 1) closest with rice cold tolerance at booting stage are presented. In parents, six morphological traits at Lijing2 had higher than Towada except BG. Among the physical attributes variational coefficient of BG was maximum (29.33%), while UIL had minimum (6.32%) in NILs population. These traits showed normal distribution by normal distribution testing (K-S-$p$ > 0.05). Skewness of RSS and 1-2 IL were negative, and others showed positive similarity to Kurtosis of BG and 1-2 IL, which were negative, and others had positive (Figure 1). Based on these findings, it is concluded that RSS, ULL, 1-2IL, RSLL, FG, and BG may be mapped with QTL.

**Table 3.** Variation from mean value (Standard Deviation, Std. D), coefficient of variation (CV), measure of symmetry (Skewness), normal distribution of data (Kurtosis) and significant differences (K-S-$p$) of six morphological traits including, rate of seed setting (RSS), full grains (FG), blighted grains (BG), uppermost leaf length (ULL), 1st and 2nd internode length (1-2 IL) and reciprocal secondary leaf length (RSLL), and correlated with cold tolerance at booting stage of NILs of japonica rice.

| Traits | Parents | | | Near Isogenic Lines (NILs) | | | | | |
|--------|--------|---------|--------------------|------------------|-------|--------------|-------|----------|----------|
| | Towada | Lijing2 | Standard Deviation | Mean | CV% | Mini/Max | K-S-$p$ | Skewness | Kurtosis |
| RSS | 0.24 | 0.94 | 0.13 | 0.54 ± 0.01 | 24.29 | 0.06/0.92 | 0.53 | −0.3 | 0.24 |
| FG | 24.6 | 123.7 | 23.16 | 83.49 ± 1.43 | 27.74 | 7.60/152.50 | 0.88 | 0.02 | 0.33 |
| BG | 80.3 | 7.5 | 20.42 | 69.63 ± 1.26 | 29.33 | 11.40/133.40 | 0.66 | 0.36 | −0.07 |
| ULL | 21.14 | 25.5 | 1.39 | 21.99 ± 0.09 | 6.32 | 18.04/25.88 | 0.83 | 0.23 | 0.07 |
| 1-2IL | 2.38 | 4.09 | 0.36 | 2.36 ± 0.02 | 15.19 | 1.33/3.45 | 0.69 | −0.15 | −0.12 |
| RSLL | 30.8 | 35.18 | 2.69 | 36.93 ± 0.17 | 7.28 | 30.87/47.58 | 0.18 | 0.44 | 0.34 |

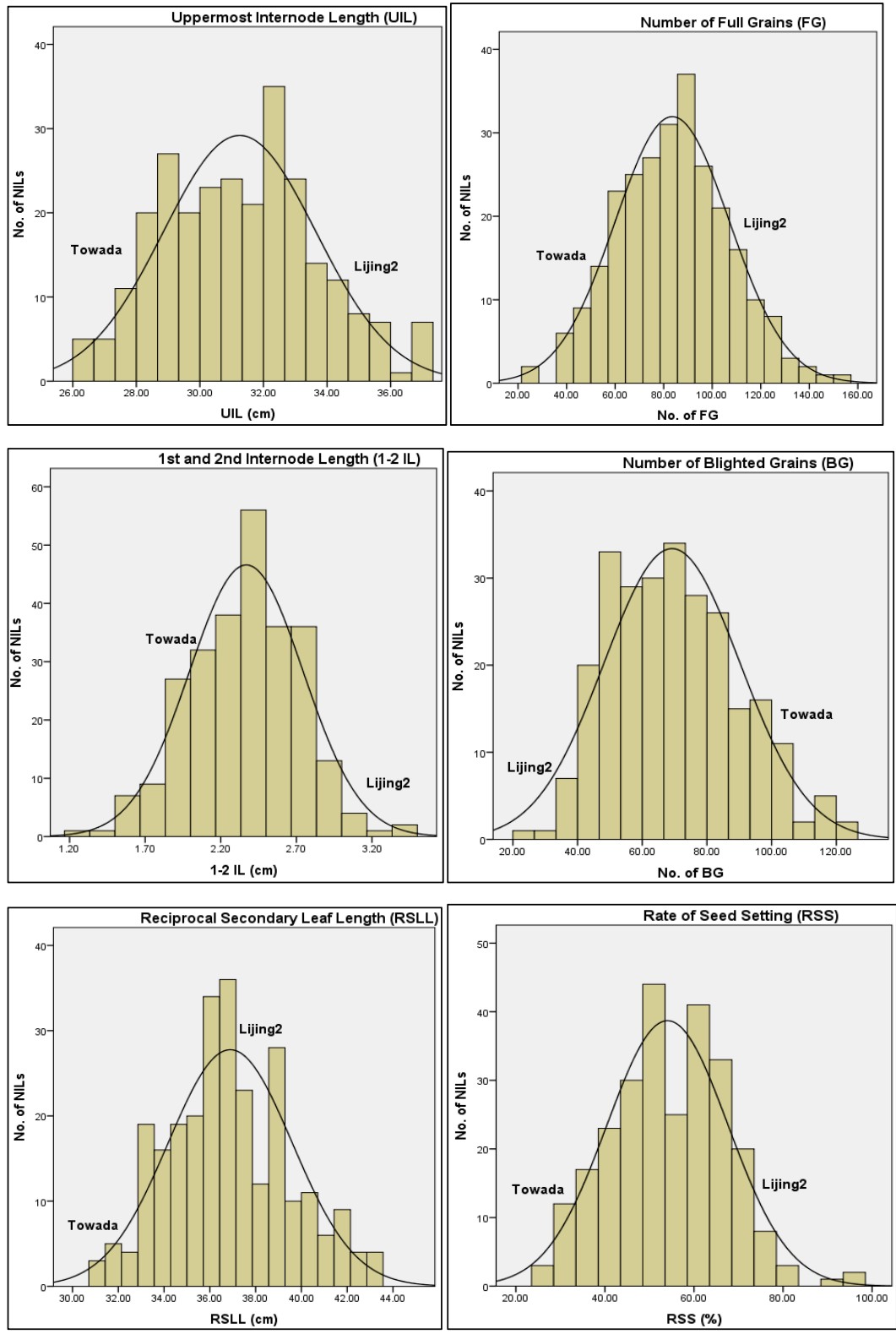

**Figure 1.** Distribution pattern of six morphological traits, full grain (FG), upper most internode length (UIL), 1st and 2nd internode length (1-2IL), blighted grains (BG), reciprocal secondary leaf length (RSLL), and rate of seed setting (RSS) in Near-isogenic lines (NILs) populations of rice plants grown under cold stress condition.

### 3.2. Polymorphism of Molecular Markers (SSR)

The initial screening of polymorphic SSR markers in the genetic pool of the parents revealed 183 differential markers out of 600, while two markers (RM3755 and RM5499), out of 183, had strong association cold tolerance, and hence these markers were used for QTL analysis. Analysis of variance (ANOVA), revealed that these two markers were significantly associated with RSS ($p < 0.01$), suggesting their strong linkage to RSS under cold stress, therefore 10 makers (RM8036, RM21473, RM21474, RM21475, RM21476, RM21478, RM21479, RM21483, RM21491, and RM7183) designed in accordance to RM3755 position in rice chromosome. Likewise, 10 makers (RM21342, RM21344, RM21345, RM6018, RM21351, RM21353, RM21354, RM5183, RM21360, and RM21361) designed according to the position of RM5499 marker. These twenty markers applied for searching QTLs on the entire $BC_5F_3$ population. ANOVA analysis of SSR markers and RSS showed RM21351, RM21361, RM5499, RM21353, RM21345, RM21344, RM21476, RM21478, RM3755, and RM8036 were significantly associated with RSS and located at chromosome no. 7 (Table 4).

**Table 4.** Simple Sequence Repeats (SSR) markers, core sequence and no. of repeats along flanking DNA sequences, used for correlation analysis of genotypic variation with phenotypic (cold tolerance) attributes at booting stage in Towada NILs.

| McCouch Locus ID | Motif | Number of Repeats | Forward Primer | Reverse Primer |
|---|---|---|---|---|
| RM21344 | AT | 12 | GGATGTTGTTCTAACCCGTCAGG | CGAACTCAACAGACTACCCATACCC |
| RM21345 | AT | 15 | GCATGCTAAGCTGTAGAAGTTAGTGG | GCTACATGTCACCGATCAGACC |
| RM21351 | AT | 15 | GGTGGTGTTTGTAAGTGTTTCACG | CGAACATGAACATAGGTCGTTGC |
| RM21353 | AT | 10 | GGAAACCACATGCTTGATGC | CACTCCTTATATGGATGGTTAGGG |
| RM21361 | AGAT | 5 | GATGGGAAGAGACGAGAGTTGACC | TAGGAGTGATACGTGGCGTCTTAGG |
| RM8036 | AT | 37 | ATGGTTTGGAGCTTGAACTGTCC | GGTTAGGAGCAGTGGGAATTGG |
| RM21476 | AGG | 10 | GACGCCGACGATCTCCATCTCC | GCAAGAGTACTATGCGGCGGAAGC |
| RM21478 | AAG | 28 | TAACACAGTTCTTCTCGCAACG | AAGTTCCCTTGTGTGATTGACC |
| RM3755 | AG | 17 | TGTGGACAACCTCAACTGAAAGC | CATAATCACCAACATCGGAGAAGC |
| RM5499 | AG | 25 | GGACGAAAGGGTATTTGATTGG | CCTCAAGGTGGTCTCCTTCTCC |

### 3.3. Linkage Mapping and Analysis of QTLs

Two QTLs (for RSS) identified (Table 5 and Figure 2), and tentatively designated as *qCTB-7-a* and *qCTB-7-b* (named referred to McCouch et al. [43]). They were located between markers RM21353–RM21345 and RM21476–RM21478 and exhibited 36% and 5.70% phenotypic variation, and −0.04% and −0.03% additive effect (from Lijiang2), respectively, while the genetic distances between the peaks of these two QTL had 0.06 cM and 0.00 cM.

**Table 5.** Marker intervals, locus distance (cM), log of odd (LOD) score, additive effect in terms of percentage for the QTLs identified on chromosome no. 7 governing rice traits associated with cold tolerance including, RSS, 1-2 IL, BG, RSLL, FG, ILBS, and ULL at the booting stage of NILs.

| Traits | Chromosome Number | Interval Distance | Site (cM) [a] | LOD | Additive Effect (%²) [b] | H²(A) % [c] |
|---|---|---|---|---|---|---|
| RSS | 7 | RM21353–RM21345 | 0.06 | 3.94 | −0.04 | 8.36 |
| RSS | 7 | RM21476–RM21478 | 0.00 | 3.03 | −0.03 | 5.7 |
| 1-2IL | 7 | RM21345–RM21344 | 0.06 | 2.60 | −0.11 | 7.31 |
| BG | 7 | RM21353–RM21345 | 0.06 | 3.60 | 5.98 | 7.67 |
| BG | 7 | RM21476–RM21478 | 0.00 | 2.06 | 4.22 | 3.92 |
| RSLL | 7 | RM21345–RM21344 | 0.48 | 2.83 | 0.65 | 5.79 |
| FG | 7 | RM21353–RM21345 | 0.04 | 2.94 | −6.22 | 6.63 |
| FG | 7 | RM3755–RM8036 | 0.00 | 2.71 | −5.45 | 5.2 |
| ILBS | 7 | RM3755–RM8036 | 0.08 | 2.03 | −0.5 | 4.37 |
| ULL | 7 | RM5499–RM21353 | 0.00 | 3.14 | −0.34 | 5.71 |
| ULL | 7 | RM21476–RM21478 | 0.00 | 3.67 | −0.38 | 6.87 |

[a] Distance on chromosome between the QTLs and the marker. [b] Addictive effect observed in parents and NIL. [c] Role of the dominance or additive effects to the morphological variation. Only $p < 0.001$ were accepted as significant for additive or dominance effects.

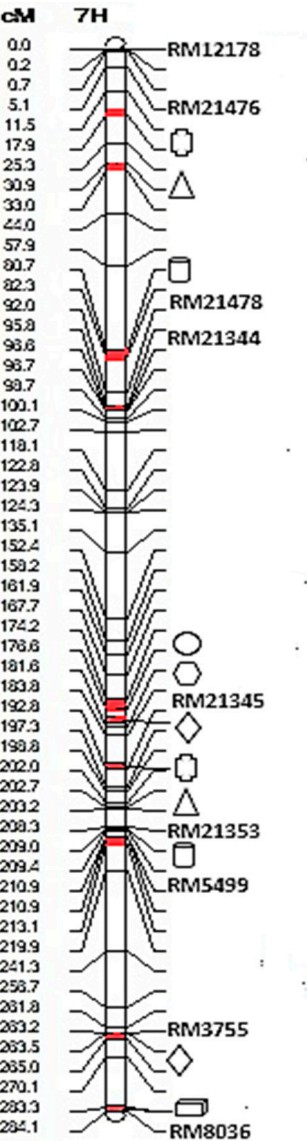

**Figure 2.** This figure shows the location and congregation of QTL affecting cold stress tolerance in NILs populations of Japonica rice. These QTL of RSS (⬡), BG (△), ILBS (▭), ULL (▢), RSLL (◯), FG (◇), and 1-2IL (◯) are located on chromosome number 7.

Moreover, another QTL identified for 1-2IL was located between makers RM21345 and RM21344 and showed 7.31% phenotypic variation. Similarly, two QTLs between makers RM21353–RM21345 and RM21476–RM21478 were attributed to BG and had 7.67% and 3.92% phenotypic variation, respectively. In addition, one QTL for RSLL also found between makers RM21345 and RM21344 with 5.79% phenotypic variation. Likewise, another two QTLs, which were located between makers RM21353–RM21345 and RM3755–RM8036 and had 6.63% and 5.2% phenotypic variation also found for FG. A QTL for ILBS was also found and located between makers RM3755–RM8036 having 4.37% phenotypic variation. Similarly, two QTLs of UIL were located between makers RM5499–RM21353 and RM21476–RM21478. All these nine QTL had additive effects from Lijing2 except BG and RSLL.

## 4. Discussion

In this study response of rice plant to chilling stress were evaluated and discerning the location of key-genes conferring cold stress tolerance at the sensitive growth (flowering) stage of rice were carried out in near-isogenic lines (NILs) of *japonica* rice. Being the model crop and prime source of calorie intake for half of the word population, improvement of rice yield and quality are hot topics and is an important but challenging task for scientific community as well as for rice breeders because plants responses for extreme temperatures is a complex phenomenon that is highly influenced by environmental factors as well as plants phenological stage [1,50] and the intensity of cold treatment (temperature, duration and recovery) is of great importance for meaningful evaluation of cold tolerance [22]. Different temperature, duration and recovery condition applied at different growth stages for rice cold stress analysis [13,30]. However, cold stress at reproductive stage is devastating and may cause male sterility, and consequently severe loss in rice quality and yield occurs [51]. Cold tolerance at this stage can be evaluated by seed setting rate, based on the cold deep-water irrigation (CDWI) system or cold greenhouse cultivation (CGC) [5,23]. Therefore, for assessment and evaluation of cold stress tolerance at booting stage of rice, this study was designed to consider both important aspects, CDWI for stress application and rate of seed setting (RSS) to evaluate cold tolerance attributes because it is most practical as well as accurate method and important attribute, respectively [19–21,52]. However, RSS can be collected after rice harvesting hence in anterior stage of rice life cycle, selection of appropriate material for cold tolerance analysis is difficult [5,53]. Therefore, identification of morphological traits especially in early stage of rice growth having strong correlation with RSS had important significance for selecting cold tolerance rice material at booting stage [5,50]. Previously, scientists attempted different studies to fulfil this task, e.g., Shen et al. [54] who analyzed correlations between 17 morphological traits with RSS through 84 NILs populations and found 11 traits, FLW, RSLL, PL, AL, AW, SIL, PHt, SL, FG, BG, and NOGP correlated with RSS at 0.01 level [52,55]. Similarly, Cui et al. [26] analyzed correlations between 16 morphological traits with RSS from 29 NILs populations and showed six traits (ILBS, FG, BG, PHt, ULL, and AL × AW) had strong correlations with RSS at 0.01 level. However, correlation between traits and RSS under natural CDWI system, followed by classification of morphological traits correlated with RSS and application in identification and characterization of molecular markers location (QTL analysis) were not performed before, and hence followed the achievement accomplished in this study.

This study revealed strong correlation of morphological attributes with the rate of seed setting (RSS), which showed that rice plants have developed well defined phenological plasticity to cope with harsh environmental conditions (Cold stress) and to manage and divert its resources to accomplish reproductive tasks [22,30,55,56]. In addition of RSS, five other phenotypic traits (ULL, 1-2IL, RSLL, FG, and BG) showed strong association with rice cold tolerance at booting stage, as observed in many other traits associated with cold tolerance [1,26]. Two QTLs for RSS were identified (Table 5 and Figure 2), and tentatively designated as *qCTB-7-a* and *qCTB-7-b* [45]), located between RM21353–RM21345 and RM21476–RM21478 exhibiting 36% and 5.70%, phenotypic variation with −0.04% and −0.03% additive effect (from Lijiang2), respectively. QTL analysis revealed five morphological traits had the same or nearer marker interval with cold tolerance, and hence considered strong evidence of cold tolerance QTLs because cold stress governing QTLs is more accurate when it showed significant and maximum association with the target (RSS) attribute [5,23]. Thus, the comparative analysis of scientific literature on this topic validated our foregoing research at molecular level and it will assist in co-evaluation of cold tolerance traits (Morphological and molecular) followed by application in modern technologies, like marker-assisted selection (MAS).

Modern techniques including MAS emerged as a powerful technique to produce and select cold tolerant lines and further application in plant breeding because it is more reliable and reproducible method. Therefore, several QTLs for low temperature stress tolerance in rice have already reported in the last two decades and identification of new markers will be carried out with the expansion of DNA markers and linkage maps. Previously several QTLs including *qCTP11*, *qCtss11* and *qCTS4a*, *qPSST-3*,

*qLTB3*, *qCTB2a*, and *Ctb1*, identified for cold tolerance in rice at germination, seedling, and reproductive growth stage, respectively [28,30,53,57], but very few of them have been successfully cloned yet [51]. Moreover, breeding of rice, engineered to cope with extreme weather conditions is highly demanded, so expansion in the elucidation of new QTLs related to chilling stress tolerance has been a noteworthy progress [5] for easier MAS in future rice breeding programs.

To fulfill this task and to contribute in achieving this goal, near-isogenic lines of two contrasting parents (Towada/Lijing2) for cold tolerance were developed because NILs are preferred materials for QTLs identification in crops and many QTLs has been mapped and cloned for crop improvement [28,50,58,59]. Owing to the importance of the potentiality of QTLs in application of modern agriculture, two new QTLs identified on chromosome number seven, and hence increased in the number of identified QTLs associated with chilling stress tolerancing in rice at booting stage [5,53,60] and comparing previously identified QTLs, the interval (between markers) and location of current loci was unique, suggesting that the findings are new and further assessment of colocalization and cloning is required. These are the fifth and sixth QTL mapped for rice cold tolerance at booting stage, but first report regarding application of cold-water irrigation system as well as in the consideration of diverse growth-related attributes with RSS. Moreover, current QTLs (*qCTB-7-a* and *qCTB-7-b*) had the biggest phenotypic variation in all QTLs at 7th chromosome at present, which can be exploit for analysis of minor QTLs with large effect and major QTLs with minor effects. To serve the precise mapping of these two QTLs, plant lines screened from segregating population are being analyzed. The marker-based selection will surely contribute to clone cold-response associated genes [5,30,51], a better understanding of the mechanism underlying the plant cold tolerance and further crop improvement.

## 5. Conclusions

Cold stress tolerance is complex but important attribute for sustainable rice breeding in times of climate change. Stress management is a difficult job for sessile creatures, and hence have diverse mechanisms of amelioration. However, different cultivars have various responses based on its ability to understand, respond, and memorize the environmental signals in harsh environmental conditions. Therefore, the performances of different cultivars of the same species are different under unfavorable conditions, because they have variations in defending challenging environment. Hence, selection of appropriate candidates for cultivation based on a firm knowledge is the resource and time saving strategy, particularly where the prevailing condition of the environment is highly variable and unpredictable. While current study aided in understanding and identifying cold tolerant rice at the flowering stage, further research is needed for implementation in MAS. Secondly, current demand is not only tolerant to stress conditions, but development of high-quality rice is also needed to cope with malnutrition issues. Therefore, our next objective is to evaluate the quality of rice grain, e.g., minerals contents, starch quality, and proteins. Thus, we can use these cold tolerant NILs and produce nutritious rice to cope with both malnutrition and hunger.

**Supplementary Materials:** The following are available online at http://www.mdpi.com/2073-4395/9/1/40/s1, Table S1: List of SSR markers.

**Author Contributions:** Conceptualization, Y.Z. and Z.S.; methodology, Z.S.; J.D.; X.P.; software, M.K.A.; validation, X.Y.; X.L.; formal analysis, X.P.; J.D.; investigation, Z.S.; resources, M.R.; C.D.; data curation, J.D.; X.P.; X.Y.; M.R.; writing—original draft preparation, Z.S. and M.K.A.; writing—review and editing, M.K.A.; visualization, C.D.; J.D.; supervision, Y.Z.; project administration, X.L.; Y.Z.; funding acquisition, Y.Z.

**Funding:** This work was funded by the National Natural Science Foundation of China (No. 31760376; 30660092), Cooperation Program between Province and Zhejiang University from Yunnan provincial Scientific and Technology Department (2006YX12) and Yunnan Introduction and Foster Talent (No. 2005PY01-14).

**Acknowledgments:** The authors thank Shuming Yang and Tao Yang for participating in some field experiments.

**Conflicts of Interest:** The authors declare no conflict of interest.

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
