# Peer review of "Near-Isogenic Lines of Japonica Rice Revealed New QTLs for Cold Tolerance at Booting Stage"

_agronomy, doi:10.3390/agronomy9010040_

Reviewer 1 Report

The report “Near-Isogenic Lines (NILs) of Japonica Rice Revealed 2 New QTLs for cold-tolerance at booting stage” describes the generation and analysis of near isogenic lines of the landrace Linjing2 (cold-tolerant) in the cold-sensitive japonica background Towada. While the approach and methods seem to be appropriate the language, especially in results and discussion section are very poor. Actually it was not possible for me to evaluate the results due to inaccuracy and grammar problems. Moreover there are many typos. Only after improving the language I am able to evaluate the data.

For introduction and methods some suggestions below:

Line 46: Replace ‘disturb’ by ‘disturbance of’

Line 47/48: Improve the sentence. What do you mean by ‘environmental orders’?

Line 51: Why past tense for ‘influenced’ and ‘affected’?

Line 66: Replace ‘depends’ by ‘depends on’.

Line 84: Replace ‘governed’ by ‘are governed’.

Line 87: Correct ‘Nils’ to NILs’.

Line 95: Replace ‘are’ by ‘is of’.

Line 96 Delete ‘and are’.

Line 104-107: Check grammar of this sentence.

Line 112-113: Check grammar of the sentence.

Line 129-130: Correct grammar and typo in the sentence.

Line 142: Provide final concentration of primers in the PCR reaction.

Author Response

Response to Reviewer 1.

 Line 46: Replaced ‘disturb’ by ‘disturbance of’

 Line 47/48: Improved the sentence.

 Line 51: Corrected

 Line 66: Replaced ‘depends’ by ‘depends on’.

 Line 84: Replaced ‘governed’ by ‘are governed’.

 Line 87: Corrected ‘Nils’ to NILs’.

 Line 95: Replaced ‘are’ by ‘is of’.

 Line 96 Deleted ‘and are’.

 Line 104-107: Checked grammar of this sentence.

 Line 112-113: Checked grammar of the sentence.

 Line 129-130: Corrected grammar and typo in the sentence.

 Line 142: Provided final concentration of primers in the PCR reaction.

Reviewer 2 Report

Minor but massive corrections are required, which are indicated directly in 'pdf' file of the manuscript. Authors have to consider each comment and make their changes accordingly to improve a revised version of the manuscript.

Author Response

Prepared according to the comments. 

Round  2

Reviewer 1 Report

The revised report by Sun et al. of which I reviewed the first version as well is scientifically mostly sound. However the language is still not acceptable, therefore a language editing service is warmly recommended to the authors. Especially in “results” almost every sentence needs to be revised because of flaws in the grammar. It is not the duty of a reviewer to correct all these errors therefore only a few examples of poor language are shown here:

Line 331-332: The detailed methods were that twenty morphological traits inKunming subtracted in Aziying then calculated the correlation.

Line 338-340: These five morphological traits maybe existed linkage with cold tolerance at booting stage and it needs to be made sure farther by molecular approach (QTL identification).

Line 385-387: Among 600 SSR markers, the parents exhibited 183 polymorphic markers and among them two  markers (RM3755 and RM5499) found as candidate markers differing cold tolerant and cold sensitive  pools hence for the entire BC5F3 population these markers used for further analysis.

These are only 3 out of many examples. My recommendation to the authors is to ask a professional language editor to correct their manuscript.

Scientifically there is the small problem that the abbreviation RSS is defined differential in the manuscript at various occasions (line 286: rate of seed setting; line 329: rice yield; line 515: seed development rate). This should be corrected that only one definition is valid for the abbreviation RSS.

Line 332-333: In the paragraph above this sentence it is mentioned that 7 traits correlated with RSS which was in the comparison between different locations. Here the comparison is between different genetic materials. To make this clearer to the reader I suggest to introduce 2 separated subparagraphs with distinct titles.

Line 340-341: I think that this is the title of a paragraph. Please introduce a blank line before.

Author Response

Line 331-332: The detailed methods were that twenty morphological traits inKunming subtracted in Aziying then calculated the correlation.

Line 338-340: These five morphological traits maybe existed linkage with cold tolerance at booting stage and it needs to be made sure farther by molecular approach (QTL identification).

Line 385-387: Among 600 SSR markers, the parents exhibited 183 polymorphic markers and among them two  markers (RM3755 and RM5499) found as candidate markers differing cold tolerant and cold sensitive  pools hence for the entire BC5F3 population these markers used for further analysis.

Answer 1. The above sentences modified in the original paper.

These are only 3 out of many examples. My recommendation to the authors is to ask a professional language editor to correct their manuscript.

Scientifically there is the small problem that the abbreviation RSS is defined differential in the manuscript at various occasions (line 286: rate of seed setting; line 329: rice yield; line 515: seed development rate). This should be corrected that only one definition is valid for the abbreviation RSS.

Answer 2. Corrected.

Line 332-333: In the paragraph above this sentence it is mentioned that 7 traits correlated with RSS which was in the comparison between different locations. Here the comparison is between different genetic materials. To make this clearer to the reader I suggest to introduce 2 separated subparagraphs with distinct titles.

Answer 3. Subparagraph introduced.

Line 340-341: I think that this is the title of a paragraph. Please introduce a blank line before.

Answer 4. Rectified.